# Predicting Biochemical Recurrence from Prostatectomy Slides - the LEOPARD Challenge

**Khrystyna Faryna**[1]
**Clément Grisi**[1]
**Nefise Uysal**[1]
[1] *Department of Pathology, Radboud University Medical Center, Nijmegen, The Netherlands*
**Jeroen van der Laak**[1,2]
**Geert Litjens**[1]
[1] *Department of Pathology, Radboud University Medical Center, Nijmegen, The Netherlands*
[2] *Center for Medical Image Science and Visualization, Linköping University, Linköping, Sweden*
**The LEOPARD Challenge Consortium**[3]
[3] *The complete list of consortium authors is provided in Appendix A.*

## Abstract

Prostate cancer is one of the most common cancers among men worldwide, and nearly one-third of patients who undergo surgical treatment experience biochemical recurrence. Current clinical tools predict risk at the population level but are less accurate for individuals. The LEarning biOchemical Prostate cAncer Recurrence from histopathology sliDes (LEOPARD) challenge benchmarked AI models to predict time to recurrence directly from H&E-stained prostatectomy slides using 2,181 cases from four countries. Top AI models achieved a C-index of 0.740, comparable to histopathological grading, C-index 0.739, and slightly below Cancer of the Prostate Risk Assessment Post-surgical (CAPRA-S), C-index 0.785. Combining AI with histopathological grading improved performance to 0.766, and with CAPRA-S to 0.799, significantly enhancing recurrence risk prediction.

**Keywords:** Prostate Cancer, Biochemical Recurrence, Deep Learning, AI, Histopathology

## 1. Introduction

Prostate cancer affects approximately 1.4 million men annually (Sung et al., 2021). After prostatectomy, PSA levels are used to monitor recurrence, typically dropping below 0.1 ng/mL within weeks (Goonewardene et al., 2014). However, nearly one-third of patients develop biochemical recurrence, linked to worse outcomes (Freedland et al., 2005; Han et al., 2001). While PSA screening is debated (Force, 2018; Heijnsdijk et al., 2018), it remains essential post-surgery. In current clinical practice, risk stratification relies on models like Cancer of the Prostate Risk Assessment (CAPRA), pre-operative (Cooperberg et al., 2005) and Cancer of the Prostate Risk Assessment Surgical (CAPRA-S), post-operative (Cooperberg et al., 2011), which combine clinical and pathological factors to group patients by risk (Cornford et al., 2024). Despite clinical value, these models are limited at the individual level due to interreader variability and potential incomplete use of available data (Epstein, 2010; Pierorazio et al., 2013).

Recently, it was shown that AI can predict biochemical recurrence directly from histopathology slides, potentially identifying morphology not incorporated into conventional grading (Pinckaers et al., 2022; Eminaga et al., 2024; Dietrich et al., 2021).

We introduce the LEarning biOchemical Prostate cAncer Recurrence from histopathology sliDes (LEOPARD) challenge to benchmark AI models for predicting time to recurrence from H&E-stained prostatectomy slides.

## 2. Materials and Methods

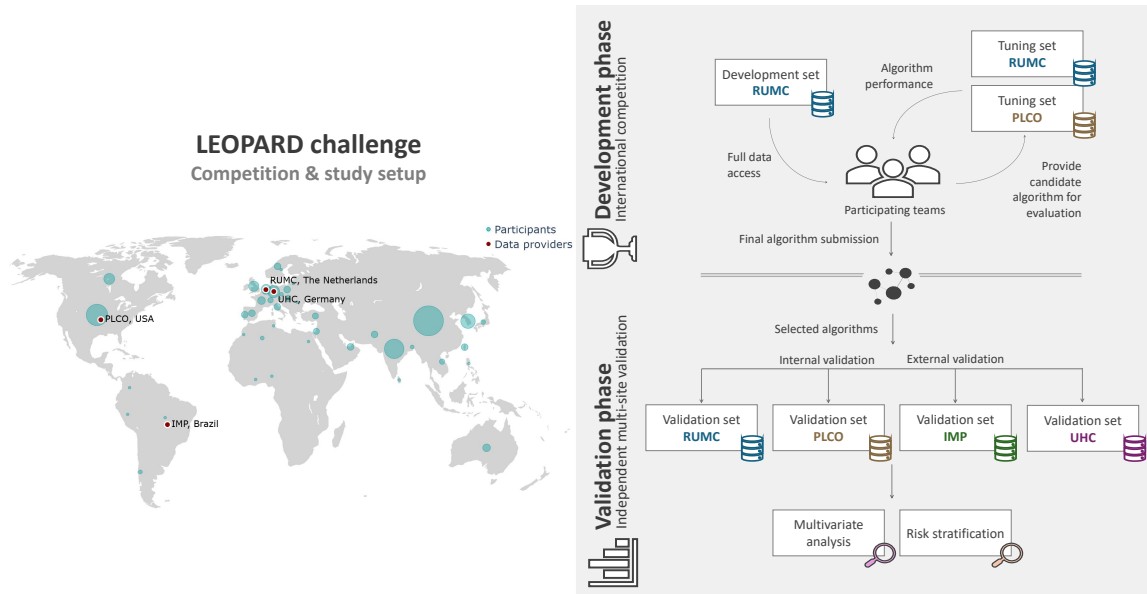

Figure 1: Distribution of > 440 AI developers (> 40 countries) participating in the LEOPARD challenge (left). Study and challenge setup (right).

We retrospectively collected whole-slide images of prostate biopsies, along with clinical and pathological data, from Radboud University Medical Center (RUMC), the Netherlands, Institute Mario Penna (IMP), Brazil, University hospital Cologne (UHC), Germany; additionally, the data from the PLCO Cancer Screening Trial was used in this study(Gohagan et al., 2000; Andriole et al., 2009). The dataset included prostatectomy slides from 2,181 patients, with relevant clinical parameters such as Gleason grade, PSA levels, TNM stage, recurrence status, and time to biochemical recurrence or last follow-up. Biochemical recurrence status was defined as a postoperative PSA rise, with center-specific thresholds: RUMC, PLCO, and IMP used $\geq$ 0.2 ng/mL confirmed by a second measurement, while UHC used a threshold of > 0.1 ng/mL following the latest European Association of Urology guidelines. The data was divided into development, tuning, and internal and external validation subsets to develop and assess predictive models.

The RUMC cohort consisted of 657 cases. These patients underwent radical prostatectomy at RUMC between 1992 and 2012. The slides were scanned with 3DHistech P1000 scanner at $0.25\mu m/pixel$ resolution. The PLCO prostatectomy cohort data were collected between November 1993 and July 2008 from ten medical centers across the USA. The final test cohort consisted of 773 patient cases. The slides were scanned at $0.25\mu m/pixel$ using a Leica scanner. The IMP cohort included data from 421 cases. These patients underwent prostatectomy at IMP, São Paulo, Brazil, in 2016. The cases were followed for 6 years, with 2022 being the last year of follow-up. The slides were scanned at resolution of $0.26\mu m/pixel$

using a Motic Easy Scan Infinity 60N scanner. The UHC dataset cohort consists of prostatectomy specimens collected between 2015 and 2020 from 330 patients treated at UHC, Cologne, Germany. Follow-up extended through 2023. The WSIs of prostate tissue specimens were scanned at $0.23 \mu m/pixel$ resolution, using the Hamamatsu Nanozoomer S360 digital slide scanner.

Participant teams were granted access to the RUMC training dataset. The AI models were designed to predict time to recurrence. During model development, teams trained their algorithms on the publicly available data and submitted their models to the evaluation platform for performance assessment. Interim performance feedback was generated using a tuning dataset consisting of prostatectomy samples from 49 RUMC patients and 50 PLCO patients.

Subsequently, a blinded Internal Validation set of 823 patients' prostatectomies from the RUMC and PLCO was used to perform the final model evaluation. The challenge was handled via https://grand-challenge.org/ platform. After Internal Validation, the top submissions participated in an External Validation phase. The models were tested on two independent datasets: one comprising prostatectomies of 330 patients from the UHC and another containing prostatectomies of 421 patients from IMP.

To assess potential added value to existing tools, models' predictions were combined with histological grading and CAPRA-S (clinical variable based on blood PSA levels, histological grading, lymph node status, seminal vesicle invasion, surgical margin status and extracapsular extension) using Cox Proportional Hazard models. The Concordance index (C-index) was used to evaluate model performance for recurrence prediction. Finally, a permutation test was used to determine whether the CAPRA-S combined with AI ensemble significantly outperformed CAPRA-S alone.

## 3. Results and Discussion

The LEOPARD challenge, Figure 1, (April-September 2024) attracted $> 440$ participants from $> 40$ countries, with $> 200$ submissions. Sixteen teams submitted final models, of which nine were selected for external evaluation. Most approaches utilized Foundational models and Multiple Instance Learning.

Top models achieved strong Internal Validation Phase performance (C-index up to 0.723 RUMC, 0.732 PLCO) and generalized well in External Validation Phase (0.773 IMP, 0.702 UHC).

The top-five algorithms were combined into an ensemble. The AI ensemble reached an overall C-index of 0.740, comparable to 0.739 histopathological grading and 0.785 CAPRA-S. Combining AI with clinical tools results in improved risk stratification performance. AI combined with histopathological grading resulted in C-index of 0.766, and AI combined with CAPRA-S resulted in C-index of 0.799, significantly outperforming CAPRA-S alone.

AI models ensemble matched traditional grading performance using only histopathology and showed consistent predictive performance across international cohorts. Combining AI with histopathological grading or CAPRA-S improved prognostic accuracy, demonstrating complementary value beyond existing clinical tools.

## Acknowledgments

We thank the Grand-Challenge team for their support during the course of the challenge.

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

## Appendix A. The Leopard Challenge Consortium

**Khrystyna Faryna**[1], **Clément Grisi**[1], **Nefise Uysal**[1], **Hans Pinckaers**[1], **Joep Bogaerts**[1], **Solène-Florence Kammerer-Jacquet**[2], **Pierre Allaume**[2], **Vittorio Agosti**[1,3], **Enrico Munari**[3,4], **Paulo Guilherme de Oliveira Salles**[5], **Yuri Tolkach**[6], **Reinhard Büttner**[6], **Sofiya Semko**[7], **Maksym Pikul**[7], **Axel Heidenreich**[7], **Adam Kowalewski**[1,8,9], **Leslie Tessier**[1,10,11], **Frédérique Meeuwsen**[1], **Jolique van Ipenburg**[1], **Till Nicke**[12], **Johannes Lotz**[12], **Jan Raphael Schäfer**[12], **Matthew McNeil**[13], **Anne L. Martel**[14], **Witali Aswolinskiy**[15], **Christian Aichmüller**[15], **Zijie Fang**[16], **Songhan Jiang**[17], **Xiangming Yan**[17], **Qi Ouyang**[18], **Yongbing Zhang**[18], **Surya Achanta**[19], **Nilanjan Chattopadhyay**[19], **Raja Fida Mohammad**[19], **Aditya Vartak**[19], **Nitin Singhal**[19], **Li Zhang**[20], **Aaron Kutzer**[20], **Narmin Ghaffari Laleh**[20,21], **Jakob Nikolas Kather**[20,21,22], **Doanh Cao Bui**[23], **Anh Tien Nguyen**[23], **Sunhong Park**[23], **Jin Tae Kwak**[23], **Suhang You**[24,25], **Sanyukta Adap**[24], **Siddhesh Thakur**[24], **Bhakti Baheti**[24,25], **Spyridon Bakas**[24,25,26], **Jeroen van der Laak**[1,27], **Geert Litjens**[1].

[1] Department of Pathology, Radboud University Medical Center, Nijmegen, The Netherlands

[2] Department of Pathology, CHU de Rennes, Rennes, France

[3] Department of Molecular and Translational Medicine, University of Brescia, Brescia, Italy

[4] Surgical Pathology Unit, Verona University Hospital Trust, Verona, Italy

[5] Anatomical Pathology Service, Instituto Mário Penna, Belo Horizonte, Brazil

[6] Institute of Pathology, University Hospital Cologne, Cologne, Germany

[7] Clinic of Urology, University Hospital Cologne, Cologne, Germany

[8] Faculty of Medicine, Bydgoszcz University of Science and Technology, Bydgoszcz, Poland

[9] Department of Tumor Pathology, Oncology Centre Prof. Franciszek Łukaszczyk Memorial

Hospital, Bydgoszcz, Poland
[10] Pathology Department, Institut du Cancer de l'Ouest, Angers, France
[11] Pathology Department, Centre hospitalier universitaire d'Angers, Angers, France
[12] Fraunhofer Institute for Digital Medicine MEVIS, Bremen, Germany
[13] Department of Medical Biophysics, University of Toronto, Toronto, Canada
[14] Sunnybrook Research Institute, Toronto, Canada
[15] PAICON GmbH, Heidelberg, Germany
[16] Tsinghua Shenzhen International Graduate School, Tsinghua University, China
[17] School of Computer Science and Technology, Harbin Institute of Technology, Shenzhen, China
[18] School of Science, Harbin Institute of Technology, Shenzhen, China
[19] AIRA Matrix, Mumbai, India
[20] Else Kroener Fresenius Center for Digital Health, TUD Dresden University of Technology, Dresden, Germany
[21] Medical Oncology, National Center for Tumor Diseases, University Hospital Heidelberg, Heidelberg, Germany
[22] Department of Medicine, University Hospital Dresden, Dresden, Germany
[23] School of Electrical Engineering, Korea University, Seoul, Republic of Korea
[24] Division of Computational Pathology, Department of Pathology & Laboratory Medicine, Indiana University School of Medicine, Indianapolis, IN, USA
[25] Indiana University Melvin and Bren Simon Comprehensive Cancer Center, Indianapolis, IN, USA
[26] Department of Computer Science, Luddy School of Informatics, Computing, and Engineering, Indiana University, Indianapolis, IN, USA
[27] Center for Medical Image Science and Visualization, Linköping University, Linköping, Sweden

