# OpenReview forum: "Predicting Biochemical Recurrence from Prostatectomy Slides - the LEOPARD Challenge"
_MIDL.io/2026/Short_Papers — MIDL 2026 - Short Papers Poster_

### Official Review · Reviewer_mXvV · 2026-05-03

**Rating:** 5
**Confidence:** 3

**Review:**

The paper is well-written, the foundations of the project are clear, important, and robust. With the large number of submissions, and the thorough data collection, it is clear that the authors can meaningfully contribute to the conference, and that spreading knowledge about the challenge would benefit the conference attendees.

**Summary:**

The paper presents the LEOPARD challenge for predicting biochemical recurrence after prostatectomy from H&E-stained slides from multiple venues. The challenge attracted 442 participants, and the best models matched traditional grading performance. The paper summarizes the main attributes and findings of the challenge, and offers an in-depth view of the current state of deep learning for their proposed task.

**Strengths:**

- The paper is well-written, and the motivation for the challenge, and hence the paper, is clear.
- The data collection is clearly very detailed and exhaustive, the 3 page limitation of the short paper format takes away from a detailed description of the data, however the online material makes this clear.
- The dataset is extremely valuable and it is very useful to bring attention to it.
- The rating of the submissions paints an informative picture of the current deep learning methods available today.

**Weaknesses:**

- If anything, I would add that the short paper has lost some relevance given that it has already concluded, however it has become more relevant in a different aspect, because now the submission results can be collected into an extensive review.

**Justification Of Rating:**

The paper is well written, the described challenge is interesting, important and the results are extremely valuable, therefore I believe the conference would benefit from accepting the submission.

---

### Decision · Program_Chairs · 2026-05-08

Accept (Poster)